# A broadly protective human monoclonal antibody targeting the sialidase activity of influenza A and B virus neuraminidases

Atsuhiro Yasuhara[1], Seiya Yamayoshi [1,2,3] ✉, Maki Kiso[1], Yuko Sakai-Tagawa[1], Moe Okuda[1] & Yoshihiro Kawaoka [1,2,3,4] ✉

Improved vaccines and antiviral agents that provide better, broader protection against seasonal and emerging influenza viruses are needed. The viral surface glycoprotein hemagglutinin (HA) is a primary target for the development of universal influenza vaccines and therapeutic antibodies. The other major surface antigen, neuraminidase (NA), has been less well studied as a potential target and fewer broadly reactive anti-NA antibodies have been identified. In this study, we isolate three human monoclonal antibodies that recognize NA from A/H1N1 subtypes, and find that one of them, clone DA03E17, binds to the NA of A/H3N2, A/H5N1, A/H7N9, B/Ancestral-lineage, B/Yamagata-lineage, and B/Victoria-lineage viruses. DA03E17 inhibits the neuraminidase activity by direct binding to the enzyme active site, and provides in vitro and in vivo protection against infection with several types of influenza virus. This clone could, therefore, be useful as a broadly protective therapeutic agent. Moreover, the neutralizing epitope of DA03E17 could be useful in the development of an NA-based universal influenza vaccine.

Seasonal influenza viruses (e.g., A/H1N1pdm09, A/H3N2, B/Yamagata-lineage, and B/Victoria-lineage virus) cause considerable morbidity and mortality worldwide. In addition, a pandemic of an emerging influenza A virus (e.g., A/H5N1 or A/H7N9) could occur, potentially claiming millions of lives. There is, therefore, an urgent need to develop vaccines or therapeutics that protect against a wide range of influenza subtypes or lineages[1]. Current seasonal influenza vaccines induce antibody responses against the major surface glycoprotein hemagglutinin (HA)[2]. Antibodies against the immunodominant globular head domain of HA are mainly elicited by these vaccines and inhibit virion binding to the cellular receptor, sialic acid. Because the antigenicity of the HA head differs among the HA subtypes and lineages, the vaccines induce a narrow range of strain-specific immune responses, resulting in their having little effect on emerging pandemic viruses[3]. Furthermore, the major antigenic sites in the HA head mutate

at a high rate due to immunological pressure and high plasticity[4], leading to a decrease in vaccine effectiveness.

Although antibodies against HA are typically considered as a correlate of protection from influenza virus infection, some studies suggest that antibodies against neuraminidase (NA), the other major glycoprotein on the viral surface, contribute to protection[5–9]. NA cleaves terminal sialic acids from N-linked glycans and this sialidase activity is essential for the release of progeny viruses from infected cells[8–10]. NA-specific antibody titers have been shown to correlate with protection against infection with various viruses, including A/H1N1[11,12] and A/H3N2[13–15] viruses. The main mechanism by which anti-NA antibodies inhibit virus propagation is through neuraminidase-inhibiting (NI) monoclonal antibodies (mAbs). These antibodies interfere with sialidase activity and restrict virus release, resulting in aggregation of progeny virions on the cell surface[3]. The NI antibodies reduce viral load

[1]Division of Virology, Department of Microbiology and Immunology, Institute of Medical Science, The University of Tokyo, Tokyo, Japan. [2]International Research Center for Infectious Diseases, Institute of Medical Science, University of Tokyo, Tokyo, Japan. [3]Research Center for Global Viral Infections, National Center for Global Health and Medicine, Tokyo, Japan. [4]Department of Pathobiological Sciences, School of Veterinary Medicine, University of Wisconsin-Madison, Madison, WI, USA. ✉e-mail: yamayo@ims.u-tokyo.ac.jp; yoshihiro.kawaoka@wisc.edu

and symptoms in infected mice, ferrets, and humans[5,6,15–18]. Although NI activity is a major mechanism of the antiviral activity of anti-NA antibodies, recent studies have reported that some NI mAbs requires Fc–FcγR interactions, which activate effector cells, such as natural killer cells, macrophages, and neutrophils, for in vivo protection[19,20].

Although influenza NAs are divided into three phylogenic groups–influenza A group 1 (N1, N4, N5, and N8 subtypes), influenza A group 2 (N2, N3, N6 N7, and N9 subtypes), and influenza B (Ancestral-, Yamagata-, and Victoria-lineages) (Fig. S1)–most of the amino acids in the active site of NA are highly conserved across all three groups[21]. Therefore, NA should be an attractive target for the development of universal vaccines and therapeutic antibodies. Although vaccine development guided by cross-reactive NI mAbs is a promising approach, a limited number of cross-reactive anti-NA mAbs have been reported to date. Of note, one paper reported three NI mAbs with cross-reactivities against both influenza A viruses (IAVs) and influenza B viruses (IBVs)[22]. One of three NI mAbs, clone 1G01, derived from A/H3N2-infected patients, directly targeted the active site of NA and inhibited several kinds of influenza A and B viruses in vitro and in vivo. Epitope footprints of such protective cross-reacting anti-NA mAbs have helped boost the development of universal NA vaccines, but there is still much work to be done.

In this study, we established a novel human mAb (DA03E17) with broad heterosubtypic binding to NAs from IAV group 1 (N1, N4, N5, N8), group 2 (N2, N3, N6, N7, N9), and IBVs. We found that this mAb recognized an epitope that differed from that recognized by clone 1G01. These results suggest that DA03E17 may be useful as a therapeutic agent and that its epitope is a potential target for a universal vaccine.

## Results

### Establishment of human monoclonal antibodies that recognize the NA of A/H1N1pdm09

We previously generated anti-NA mAbs from patients who were infected with A/H1N1pdm09 virus by using the hybridoma method[20]. In this study, we looked for novel cross-reactive anti-NA mAbs by re-screening. Briefly, the hybridomas were generated by fusing SPYMEG cells and peripheral blood mononuclear cells (PBMCs), which were obtained from human volunteers who were infected with A/H1N1pdm09 virus in the 2015–2016 influenza season. After several rounds of screening by ELISA using purified A/Yokohama/94/2015 virus (A/H1N1pdm09) that was isolated in the 2015–2016 season, hybridomas in the positive wells were biologically cloned. Of 701 hybridomas screened, we obtained three monoclonal antibodies (HP02A67, DA03E17, and DA05A30) that recognized the NA of A/Yokohama/94/2015 virus. We analyzed the nucleotide sequences of the variable regions of each antibody and found that the three mAbs used different VH and VL genes (Table S1).

### Binding breadth of the three anti-NA mAbs

To determine the breadth of mAb recognition, we performed an ELISA using NA-displaying virus-like particles (NA-VLPs), which were produced by expressing Ebola virus VP40 with NA derived from IAV group 1 (A/H1N1pre2009, A/H1N1pdm09, A/H5N1, A/H8N4, A/H12N5, A/H15N5, A/H5N8, A/H10N8), group 2 (A/H3N2, A/H7N2, A/H9N2, A/H6N3, A/H4N6, A/H5N6, A/H7N7, A/H10N7, A/H7N9), or IBV (B/Ancestral-lineage, B/Yamagata-lineage, B/Victoria-lineage) (blue in Fig. 1A and B, and Table S2). HP02A67 and DA05A30 are bound to NA-VLPs of A/H1N1pre2009 or A/H1N1pdm09, but not to those of other NAs. DA03E17 displayed broad binding to NA-VLPs of all IAVs and IBVs tested. The broadly reactive anti-NA mAb 1G01[22] showed broad binding to NA-VLPs of IAVs except for A/Puerto Rico/8/34, and relatively weak binding to NA-VLPs of IBVs. A negative control, anti-B-HA mAb 1430E3/9[20], did not bind to any NA-VLPs (Table S2). These results indicate that DA03E17

is a broadly reactive human anti-NA mAb with a different binding pattern from that previously reported for the broadly reactive clone 1G01.

### NI activity and in vitro neutralizing potency of the three mAbs

We next examined the functional capacity of the three anti-NA mAbs. NI activity is caused by direct binding to the enzyme's active site or through steric hindrance due to antibody binding at a location other than the active site. An enzyme-linked lectin assay (ELLA) can measure NI activity via either mechanism; therefore, we conducted an ELLA using NA-VLPs of IAVs and IBVs (red in Fig. 1A and B, and Table S3). HP02A67, DA05A30, and 1430E3/9 did not inhibit the sialidase activity of any of the NA-VLPs tested. In contrast, DA03E17 inhibited the sialidase activity of NA-VLPs of IAVs and IBVs except for A/mallard/Sweden/24/2002 (A/H8N4) at $IC_{50}$ values of between 0.002 and 47.1 μg/ml. 1G01 inhibited the sialidase activity of NA-VLPs of IAVs and IBVs at $IC_{50}$ values of 0.012–40.3 μg/ml, except for A/Puerto Rico/8/34 (A/H1N1), A/mallard/Interior Alaska/7MP0167/2007 (A/H12N5), A/mallard duck/Sweden/139579/2012 (A/H15N5), B/Phuket/3073/2013 (B/Yamagata-lineage), and B/Texas/02/2013 (B/Victoria-lineage).

Next, to determine whether the NI activity of DA03E17 is caused by direct inhibition, we performed an NA-Star assay, which utilizes small molecules as a substrate (Table 1). DA03E17 inhibited the sialidase activity of all of the NA-VLPs tested ($IC_{50} = 0.87$–20.7 μg/ml), whereas 1G01 failed to inhibit the sialidase activity of NA-VLP of B/Phuket/3073/2013 and B/Texas/02/2013 (Table 1). These results suggest that DA03E17 directly recognizes the enzymatic active site of NA.

To further confirm the NI activity of the three mAbs in vitro, we investigated whether these mAbs had in vitro neutralizing potency in a microneutralization (MN) assay using authentic viruses (green in Fig. 1 and Table S4). HP02A67 and DA05A30 failed to neutralize any viruses tested even at the highest concentration (50 μg/ml). DA03E17 neutralized both IAVs and IBVs except for A/England/261/91, B/Lee/40, and B/Wisconsin/01/2010. 1G01 neutralized IAVs except for A/Puerto Rico/8/34; weakly neutralized B/Lee/40, B/Yamagata/1/73, and B/Colorado/06/2017; and did not inhibit the other four IBVs tested. 1430E3/9 and anti-A/H1N1-HA mAb F3A19[23], which served as negative controls for IAVs and IBVs, respectively, did not inhibit virus propagation. Taken together, these results show that DA03E17 possesses more broad reactivity and inhibition potency than 1G01.

### In vivo protective efficacy of DA03E17 against lethal challenge infection

To evaluate the in vivo protective efficacy of DA03E17, we examined whether this mAb protected mice from lethal challenge with A/H1N1pdm09, A/H5N1, A/H3N2, A/H7N9, or B/Yamagata virus. Four mice per group were intraperitoneally inoculated with DA03E17, 1G01, the negative control mAb (1430E3/9 or F3A19 for IAVs or an IBV, respectively) at 10, 2, or 0.4 mg/kg before lethal challenge infection and their body weight changes and survival were monitored for 14 days. When mice received each mAb at 10 mg/kg, DA03E17 completely or partially protected mice from A/H1N1pdm09, A/H5N1, A/H3N2, A/H7N9, or B/Yamagata lethal challenge infection (red line in Fig. 2A). 1G01 also protected mice from A/H3N2, A/H1N1pdm09, A/H5N1, or B/Yamagata challenge, but failed to protect any mice from A/H7N9 challenge (blue line in Fig. 2A). All mice that received 10 mg/kg of 1430E3/9 or F3A19 died within 10 days of inoculation (gray line in Fig. 2A). At the 2 mg/kg dose, DA03E17 conferred full or partial protection against A/H1N1pdm09, A/H5N1, A/H3N2, and B/Yamagata virus infection with severe or moderate body weight loss, but no mice that were challenged with A/H7N9 virus survived (red line in Fig. S2A). At the 0.4 mg/kg dose, DA03E17 protected one mouse from the H5N1 virus challenge (red line in Fig. S2B). These results indicate that DA03E17 possesses in vivo antiviral activity against several subtypes of influenza virus.

We also assessed virus titers at 2 and 4 days post-infection in the lungs of infected mice that received each mAb at 10 mg/kg (Fig. 2B).

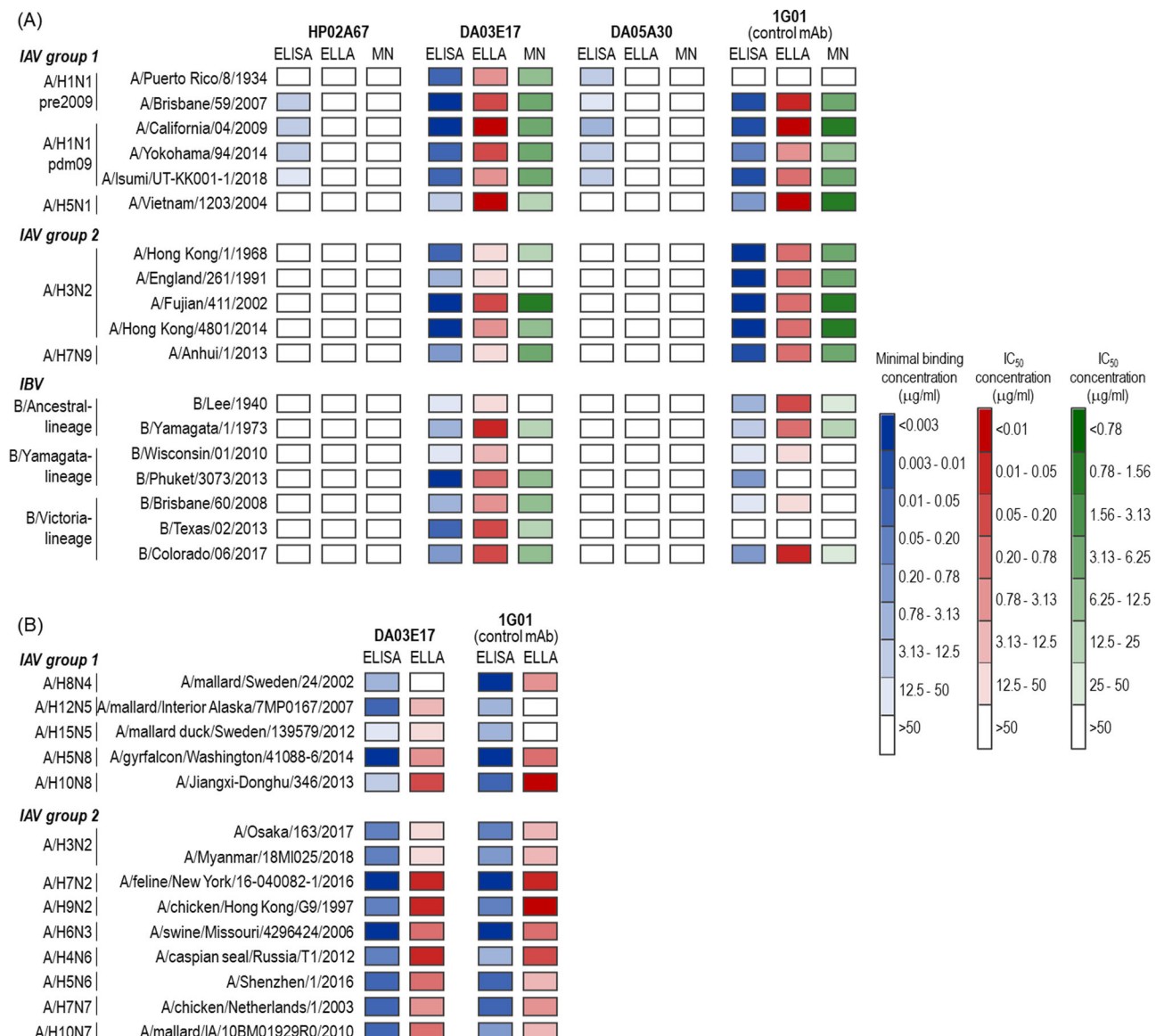

**Fig. 1 | Cross-reactivity of the three mAbs.** Heat maps of antibody binding to NA-displaying VLPs in ELISA, NI activity in ELLA, and neutralization potencies in an MN assay are shown in blue, red, and green, respectively. NAs are derived from **A** circulating human influenza subtypes or highly pathogenic viruses (A/H5N1 or A/ H7N9) or **B** recent A/H3N2 human isolates or animal influenza viruses. The exact minimal binding concentration, 50% inhibition concentration for sialidase activity ($IC_{50}$), and 50% neutralization concentration are listed in Tables S2–S4. Source data are provided as a Source Data file.

On day 2 post-infection, virus titers were not affected by the administration of any mAb. On day 4 post-infection, virus titers in the lungs of infected mice that received DA03E17 or 1G01 were significantly lower than those in the lungs of mice that received the negative control

1430E3/9. These results demonstrate that DA03E17 directly suppresses virus propagation following lethal infection.

### Mutations that permit escape from DA03E17
To identify the epitope of DA03E17, we attempted to select escape mutant viruses in triplicate (designated lines 1, 2, and 3) by passaging A/H3N2 virus (A/Hong Kong/4801/2014) in the presence of two-fold serially diluted DA03E17 (50–0.78 μg/ml). Mutant viruses that escaped from DA03E17 emerged by 4–6 passages (Table 2). To identify the escape mutation, we analyzed the nucleotide sequences of the HA, NA, and six other segments by direct sequencing. We found that escape mutant line 1 possessed the D151G substitution in NA, line 2 possessed the D151N and T439A substitutions in NA, and line 3 possessed the V50L and D151G substitutions in NA (Table 2). In the HA segment, the S110L and P221H substitutions were found in lines 1–3, respectively. No amino acid substitutions occurred in the other six segments.

### Table 1 | IC₅₀ value (μg/ml) of neuraminidase inhibition activity by mAbs in a small substrate NA-Star assay

| Subtype | Origin of NA used | DA03E17 | 1G01 | 1430E3/9 |
|---|---|---|---|---|
| A/H1N1pdm09 | A/California/04/2009 | 1.06 | 1.19 | >50 |
| A/H5N1 | A/Vietnam/1203/2004 | 0.87 | 0.83 | >50 |
| A/H3N2 | A/Hong Kong/4801/2014 | 11.7 | 20.0 | >50 |
| A/H7N9 | A/Anhui/1/2013 | 11.1 | 2.83 | >50 |
| B/Yamagata-lineage | B/Phuket/3073/2013 | 20.7 | >50 | >50 |
| B/Victoria-lineage | B/Texas/02/2013 | 5.40 | >50 | >50 |

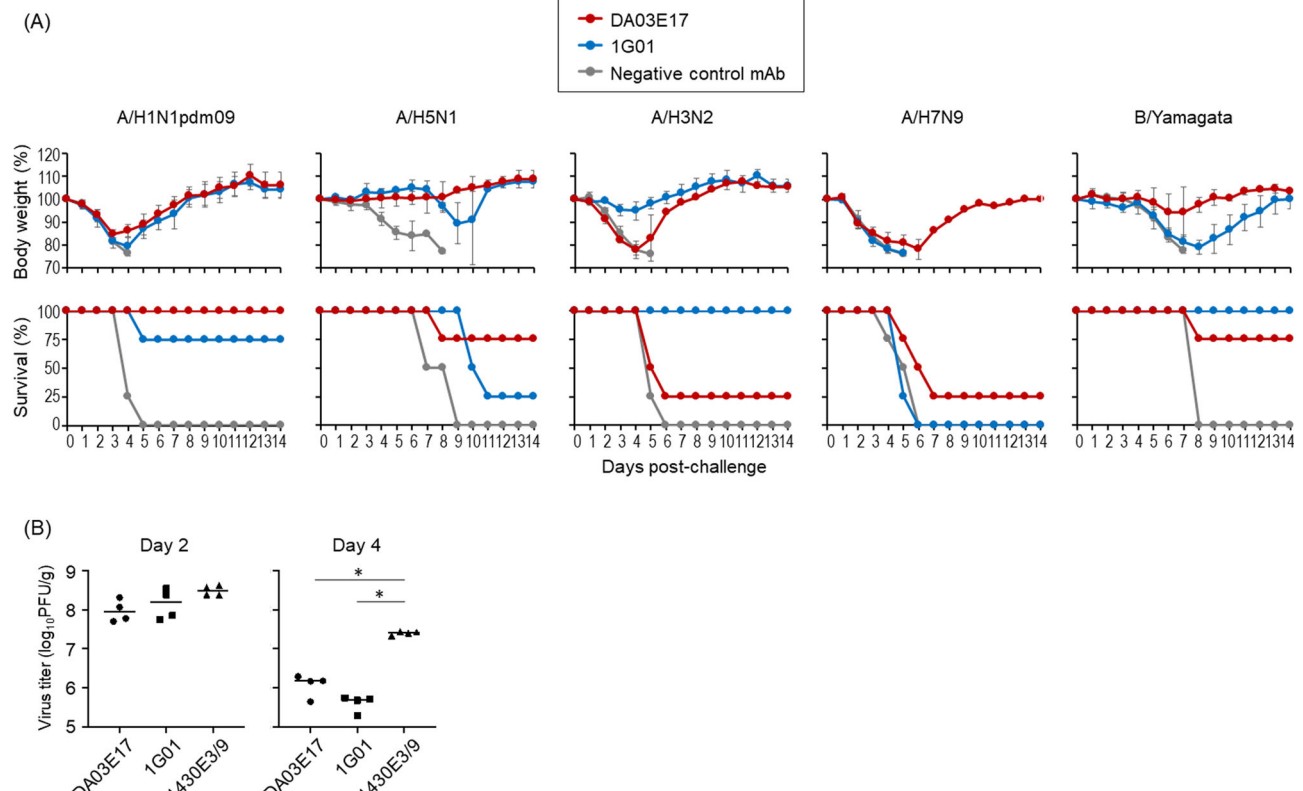

**Fig. 2 | In vivo protective efficacy of DA03E17. A** Four mice per group were intraperitoneally inoculated with DA03E17 (red line), 1G01 (blue line), or the negative control mAb (gray line) at 10 mg/kg; 1430E3/9 and F3A19 were used as negative controls for IAVs and an IBV, respectively. One day later, the mice were challenged with 10 $MLD_{50}$ of the indicated challenge viruses. Body weight changes and survival were monitored daily for 14 days. Body weight changes are shown as the mean ± SDs. **B** Four mice per group were intraperitoneally injected with the indicated antibodies at 10 mg/kg. One day later, the mice were challenged with 10 $MLD_{50}$ of the A/H1N1pdm09 virus. On days 2 and 4 post-infection, virus titers in the lungs were determined. Source data are provided as a Source Data file. *$P < 0.01$ (one-way ANOVA followed by Dunnett's tests).

To identify the key substitution for escape from DA03E17, we generated viruses possessing the D151G, D151N, and T439A, or V50L and D151G substitutions in NA by use of reverse genetics and performed the microneutralization assay. Mutant viruses possessing the D151G (found in line 1), D151N + T439A (line 2), or V50L + D151G (line 3) substitutions in NA grew well even in the presence of DA03E17 at 50 µg/ml (Table 3), indicating that these mutations were important for escape from the mAb. To pin down the amino acid substitution for DA03E17 escape, we prepared viruses possessing single amino acid substitutions (D151N, T439A, or V50L) and examined the neutralization by DA03E17. DA03E17 failed to neutralize the virus possessing the D151N substitution in NA (Table 3). The $IC_{50}$ value of DA03E17 against the virus possessing the T439A substitution was approximately twice as high as that of the

wild-type virus (Table 3). These results demonstrate that amino acid substitutions at positions 151 and 439 play a central role in escape from DA03E17. Of note, the $IC_{50}$ values of 1G01 for variants containing the D151N or D151G substitution were slightly higher than those for the wild-type virus, whereas the $IC_{50}$ value for the T439A variant was comparable to that for the wild-type virus. These results suggest that the footprint of DA03E17 on the NA enzymatic site slightly differs from that of 1G01.

To evaluate whether the epitope of DA03E17 overlaps with that of 1G01, we performed a competitive binding assay. To avoid recognition by the HRP-conjugated secondary antibody, the Fc region of 1G01 or DA03E17 was replaced with that of mouse IgG2a. Dilutions (0.006–100 µg/ml) of 1G01 with the mouse Fc domain (Fig. S3, left panel) or DA03E17 with the mouse Fc domain (right panel) were reacted with VLPs displaying NA of A/California/04/2009 (A/H1N1pdm09) and then mixed with DA03E17, 1G01, or DA05A30 (at concentrations adjusted to achieve $OD_{450}$ value = 1.5). The $OD_{450}$ value of the human Fc antibody in the presence of each concentration of mouse Fc antibody was then measured. We found that both mouse Fc antibodies showed moderate inhibition of the binding of human DA05A30, which does not possess NI activity. The mouse Fc 1G01 significantly inhibited the antibody binding of DA03E17, and this inhibition was comparable to the homologous competition (i.e., human Fc 1G01 vs. mouse Fc 1G01) (Fig. S3 left). However, in the presence of mouse Fc DA03E17 (Fig. S3 right), the mouse Fc DA03E17 inhibited binding of 1G01 but this inhibition was significantly milder than the homologous competition (i.e., human

**Table 2 | Amino acid substitutions of viruses passaged in the presence of DA03E17**

| Virus line[a] | Passage number[b] | Amino acid substitution(s) in[c] | |
|---|---|---|---|
| | | NA[d] | HA[e] |
| 1 | 4 | D151G | S110L |
| 2 | 6 | D151N + T439A | P221H |
| 3 | 6 | V50L + D151G | P221H |

[a]Lines 1–3 were independently obtained.
[b]The passage number at which escape mutants were obtained.
[c]No mutation occurred in other segments.
[d]N2 numbering.
[e]H3 numbering.

**Table 3 | Neutralization activity against mutant viruses possessing amino acid substitutions in NA**

| mAb | IC$_{50}$ value (µg/ml) against variant viruses possessing amino acid substitutions of escape mutants | | | | | | |
|---|---|---|---|---|---|---|---|
| | Wild-type | D151G | D151N + T439A | V50L + D151G | D151N | T439A | V50L |
| DA03E17 | 15.8 | >50 | >50 | >50 | >50 | 39.7 | 17.7 |
| 1G01 | 3.13 | 15.8 | 9.92 | 8.84 | 9.92 | 4.42 | 6.25 |
| CR9114 | 3.13 | 6.25 | 4.96 | 4.96 | 1.56 | 4.42 | 6.25 |
| 1430E3/9 | >50 | >50 | >50 | >50 | >50 | >50 | >50 |

DA03E17 vs. mouse DA03E17). These results suggest that the epitopes of DA03E17 and 1G01 mostly overlap, but not completely.

To visualize the antibody recognition site, we mapped these residues on the NA molecule. The amino acids at positions 151 and 439 are located in the 150-loop and the 430-loop of the enzymatic active site, respectively (Fig. 3A). D151 is conserved among all of the viruses tested and T439 is found in all of the IAVs tested (Fig. 3B). These data suggest that DA03E17 targets the highly conserved residues in the enzymatic active site of NA.

### Growth kinetics of the escape mutant viruses in vitro

To examine the fitness of the viruses possessing mutations that allowed them to escape from DA03E17, we compared the growth kinetics of mutant viruses possessing the escape mutations (i.e., D151G or D151N) in hCK cells with those of the wild-type virus. All mutant viruses tested replicated to significantly lower titers than the wild-type virus (Fig. 4). These results demonstrate that the fitness of the escape mutants was reduced.

### Characterization of the germline antibody of DA03E17

We prepared the germline mAb of DA03E17 to investigate the changes in the binding properties to NA-VLPs due to somatic hypermutation. The ELISA using NA-VLPs revealed that the germline mAb of DA03E17 recognized NA-VLPs of IAVs and IBVs, but its binding affinity was clearly lower than that of DA03E17 (Table 4). The ELLA showed that the NI activity of the germline DA03E17 was lower than that of DA03E17 and it did not cross-react with the NA of IBVs. These results suggest that affinity maturation through somatic hypermutations in the VH and VL genes plays an essential role in the high binding affinity and broad reactivity of DA03E17.

## Discussion

Novel vaccines against influenza A and B viruses should provide broad protection against diverse strains and subtypes. The development of such vaccines is aided by the identification of novel broadly neutralizing epitopes targeted by cross-reactive antibodies. Although previous studies have reported monoclonal antibodies that possess cross-reactivity against viruses belonging to the same type (IAV or IBV)[8,9,24–26], there has been only one previous report of broadly reactive antibodies against both IAV-NAs and IBV-NAs[22]. In this study, we characterized three human monoclonal antibodies derived from influenza patients. DA03E17 showed broad reactivities against NAs derived from antigenically distinct subtypes of IAVs and lineages of IBVs, whereas the other two clones (HA02A67 and DA05A30) specifically recognized the NA of A/H1N1pre2009 and A/H1N1pdm09 viruses. DA03E17 efficiently inhibited the sialidase activity of NA, whereas both HA02A67 and DA05A30 did not. Furthermore, DA03E17 showed broader cross-reactivity and better neutralization capability than 1G01, an NI mAb previously reported to have cross-reactivity against IAVs and IBVs. Moreover, our antibody protected mice from lethal challenges with several subtypes of IAVs or IBVs, suggesting that DA03E17 has potential as a therapeutic agent against both IAVs and IBVs. Although escape mutant viruses emerged in in vitro experiments, these escape mutant viruses replicated significantly less efficiently in vitro, indicating that mutant viruses that are able to escape from

DA03E17 would be unlikely to dominate the parental viruses. Given that the potential emergence of escape mutant viruses is one of the main disadvantages of mAbs as antiviral treatments, DA03E17 should be useful as a protective antibody. Furthermore, fortunately, amino acid mutations that reduced the neutralizing activity of DA03E17 had little effect on the neutralizing capability of 1G01, meaning that combination therapy of DA03E17 and 1G01 would likely suppress the emergence of escape mutants. Further studies on such combination therapy are required to develop effective, long-lasting therapies.

Currently licensed influenza vaccines, such as inactivated vaccines and live-attenuated vaccines, elicit limited anti-NA immunity[10]. Recombinant NA and NA-VLPs have been shown to induce high titers of anti-NA antibodies that protect mice and ferrets from challenge infections[27–29]. Furthermore, recombinant NA-based vaccines have been shown to inhibit influenza virus transmission in the guinea pig model[30]. In these studies, heterosubtypic immunity was not detected, although cross-protection against viruses belonging to the same subtype was observed[27]. To add heterosubtypic immunity to NA-based vaccines, antibody-targeting to conserved neutralizing epitopes is a valid approach. Therefore, the conserved epitope identified in this study might be of value in the development of NA-based universal vaccines. Furthermore, since the germline sequence of DA03E17 showed broad cross-reactivity, the epitope of this mAb could be a universal vaccine antigen that activates specific germline-precursor B cells showing broad and potent activity.

NA is a valid drug target because several small molecules that inhibit its sialidase activity (NA inhibitors) are widely approved as therapeutic agents for influenza virus infection[31]. However, several in vitro studies have shown that IBVs are less sensitive than IAVs to FDA-approved NA inhibitors and cap-dependent endonuclease inhibitors[32,33]. Therefore, novel therapeutic agents are needed. In addition to seasonal influenza viruses, A/H5N1 and A/H7N9 viruses with an amino acid substitution in NA that confers high-level resistance to NA inhibitors have been isolated from patients who were treated with NA inhibitors[34,35]. Furthermore, such NA inhibitor-resistant mutations were found in A/H5N1 and A/H7N9 viruses that were detected in poultry and the environment[36,37]. Since DA03E17 showed both broad reactivity and strong in vivo protective activity, it has the potential to be used as a therapeutic agent against seasonal and emerging influenza viruses. Further studies to develop this mAb as a therapeutic agent are warranted.

Amino acid substitution at position 151 of NA, which is involved in the escape from DA03E17, is known to confer binding to sialic acid to NA and reduce sensitivities to NA inhibitors[38–42]. Although the amino acid at position 151 frequently changed after passages in culture cells[43], aspartic acid at position 151 and threonine at position 439 are highly conserved in the NA of human isolates available in the GISAID database (https://www.gisaid.org/) (Table S5). These data suggest that the amino acids at positions 151 and 439 have functional and/or structural importance for virus replication in the human upper respiratory tract, signifying that DA03E17 has the potential to be an anti-influenza agent with a low propensity for the development of resistant viruses in humans.

The NA of the circulating human A/H3N2 viruses has undergone appreciable antigenic drift since 2016. This antigenic drift, which was

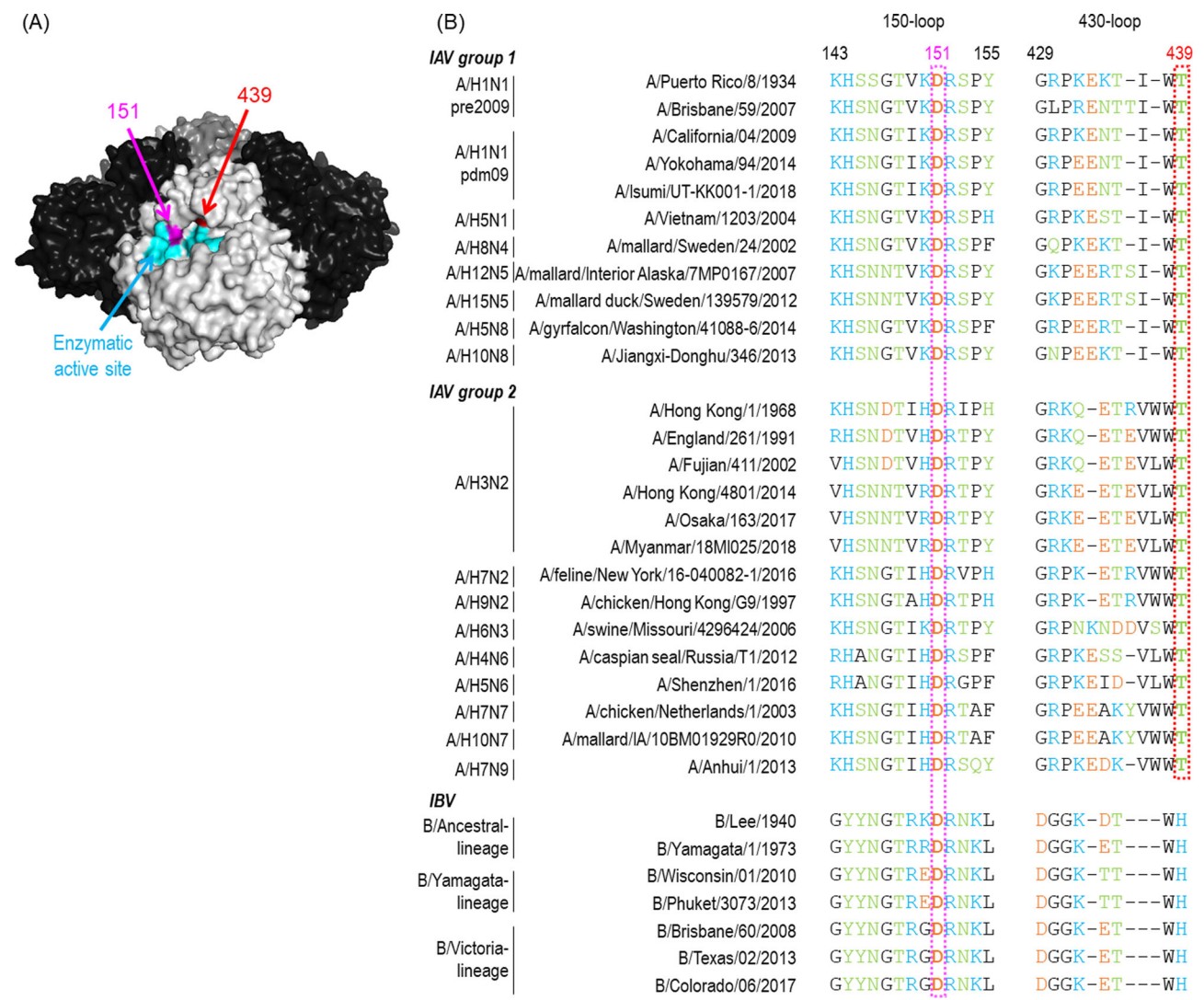

**Fig. 3 | Positions and conservation of key residues for escape from DA03E17.**
**A** Amino acid positions that are important for escape from DA03E17 are shown on the NA structure of A/Minnesota/11/2010 (A/H3N2; PDB accession code 6N6B) virus. Each NA monomer is indicated in white, gray, or black. Cyan indicates catalytic residues; magenta and red indicate amino acids that are essential for escape from DA03E17. **B** Amino acid alignment of NAs from viruses was used in this study. Amino acids with electrically positive-charged side chains, negative-charged side chains, polar side chains, or nonpolar side chains are indicated in cyan, orange, green, or black, respectively. Key residues for escape are indicated in dotted boxes in magenta and red. Source data are provided as a Source Data file.

caused by the addition of an N-linked glycosylation site near the enzymatic site, abolished the binding activity of some anti-N2-NA broadly reactive mAbs (e.g., clone B10)[44]. However, our experiments showed that DA03E17 and 1G01 maintained inhibitory activity against the NAs of recently circulating A/H3N2 viruses (Fig. 1B). This might be due to differences in the assays or cells used for the experiments because N-linked glycosylation varies among the cell lines and animal species used for virus and VLP preparation. DA03E17 and 1G01 may inhibit the NA activity of recent A/H3N2 viruses due to their structural features; we note that DA03E17 and 1G01 have longer than average CDR-H3. The average CDR-H3 length is 16 amino acids (11 amino acids for clone B10), whereas the CDR-H3 length for DA03E17 and 1G01 is 19 and 23 amino acids, respectively[45,46]. The long CDR-H3 protrudes deeper into the active pocket and forms a more extensive polar interaction network with NA, allowing its higher binding affinity[25]. Therefore, the long CDR-H3 may be a factor in allowing DA03E17 and 1G01 to bind to the NAs of recent A/H3N2 viruses, and an important feature of mAbs that broadly block NA enzymatic activity.

DA03E17 inhibited the enzymatic activity of NA derived from IAVs or IBVs, making this mAb a promising candidate for therapeutic

development. Our study suggests that DA03E17 is a potent inhibitor of NA activity in vitro and provides broad protection from mortality and morbidity in vivo. Knowledge of its epitope will be useful in the development of NA-based universal influenza virus vaccines.

## Methods

### Ethics
Human blood was collected from three volunteers by following a protocol approved by the Research Ethics Review Committee of the Institute of Medical Science, the University of Tokyo. Written informed consent was obtained from each participant. All experiments with mice were performed in accordance with the University of Tokyo's Regulations for Animal Care and Use and were approved by the Animal Experiment Committee of the Institute of Medical Science, the University of Tokyo.

### Cells
Humanized Madin–Darby canine kidney (hCK) cells were maintained in Eagle's minimal essential medium (MEM) containing 5% newborn calf serum (NCS)[43]. Human embryonic kidney 293T cells were

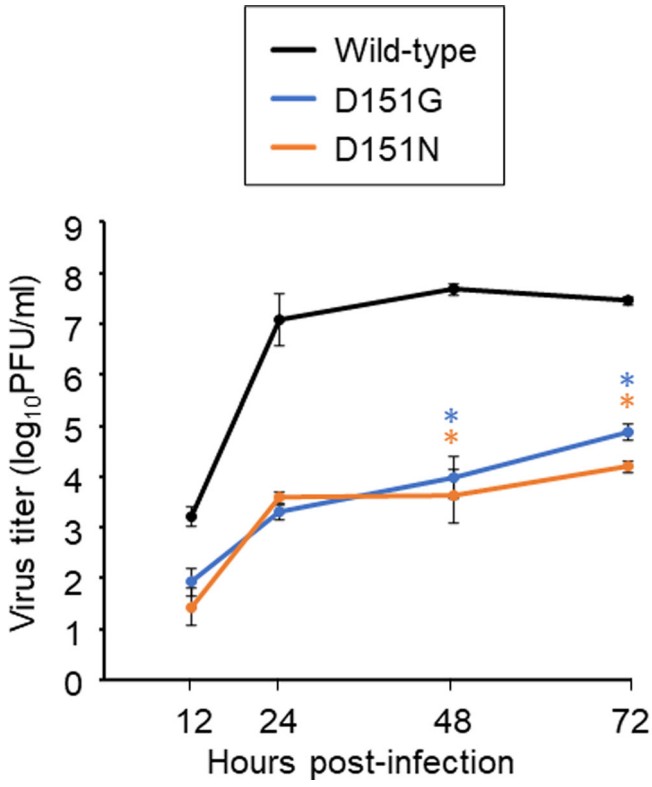

**Fig. 4 | Growth kinetics of wild-type and escape mutant viruses in vitro.** The growth kinetics of the wild-type virus and the indicated escape mutant viruses in hCK cells were compared. Cell culture supernatants of hCK cells infected at an MOI of 0.0001 were collected at 12, 24, 48, and 72 h post-infection. Virus titers are presented as the mean ± SD ($n$ = three independent experiments). Source data are provided as a Source Data file. *$P < 0.01$ (two-way ANOVA followed by Dunnett's tests).

**Table 4 | In vitro properties of DA03E17 germline mAb**

| Subtype | Origin of NA tested | ELISA[a] | ELLA[b] |
|---|---|---|---|
| A/H1N1 pre2009 | A/Puerto Rico/8/34 | 3.13 | 9.58 |
| | A/Brisbane/59/2007 | 0.0122 | 0.200 |
| A/H1N1 pdm09 | A/California/04/2009 | 0.0488 | 0.719 |
| | A/Yokohama/94/2015 | 0.781 | 2.53 |
| | A/Isumi/UT-KK001-1/2018 | 12.5 | 23.4 |
| A/H5N1 | A/Vietnam/1203/2004 | >50 | 4.64 |
| A/H3N2 | A/Hong Kong/1/68 | 0.0122 | 40.1 |
| | A/England/261/91 | 0.781 | >50 |
| | A/Fujian/411/2002 | <0.003 | 2.65 |
| | A/Hong Kong/4801/2014 | 0.0488 | >50 |
| A/H7N9 | A/Anhui/1/2013 | 12.5 | >50 |
| B/Ancestral-lineage | B/Lee/40 | >50 | >50 |
| | B/Yamagata/1/73 | 3.13 | >50 |
| B/Yamagata-lineage | B/Wisconsin/01/2010 | >50 | >50 |
| | B/Phuket/3073/2013 | 0.0122 | >50 |
| B/Victoria-lineage | B/Brisbane/60/2008 | >50 | >50 |
| | B/Texas/02/2013 | 12.5 | >50 |
| | B/Colorado/06/2017 | 50 | >50 |

[a]Minimal binding concentration (µg/ml) is shown.
[b]IC$_{50}$ value (µg/ml) is shown.

maintained in Dulbecco's modified Eagle's medium (DMEM) containing 10% fetal calf serum (FCS). SPYMEG cells (MBL) were maintained in DMEM containing 15% FCS[47]. These cells were incubated at 37 °C under 5% $CO_2$. Expi293F cells (Thermo Fisher Scientific) were maintained in Expi293 Expression Medium (Thermo Fisher Scientific) at 37 °C under 8% $CO_2$.

### Animals
Six-week-old female BALB/c or DBA2 mice (Japan SLC) were used in the study. The relative humidity was kept at 45–65%. Mouse rooms and cages were kept at a temperature range of 20–24 °C, and were set at a 12 h light:12 h dark cycle.

### Viruses
We used two A/H1N1pre2009 viruses (A/Puerto Rico/8/34 and A/Brisbane/59/2007), four A/H1N1pdm09 viruses [A/California/04/2009, A/Yokohama/94/2015, A/Isumi/UT-KK001-1/2018, and mouse-adapted A/California/04/2009[48]], an A/H5N1 virus (A/Vietnam/1203/2004), five A/H3N2 viruses [A/Hong Kong/1/68, A/England/261/91, A/Fujian/411/2002, A/Hong Kong/4801/2014, and mouse-adapted A/Aichi/2/68[49]], an A/H7N9 virus (A/Anhui/1/2013), two B/Ancestral-lineage viruses (B/Lee/40 and B/Yamagata/1/73), two B/Yamagata-lineage viruses (B/Wisconsin/01/2010 and B/Phuket/3073/2013), and three B/Victoria-lineage viruses (B/Brisbane/60/2008, B/Texas/02/2013, and B/Colorado/06/2017). All of these viruses were propagated in hCK cells. Mutant viruses were generated by the use of reverse genetics as described below.

### Plasmid-based reverse genetics
The NA mutant viruses were generated by plasmid-based reverse genetics, as described previously[50]. The pPol I plasmids encoding HA and wild-type or mutated NA derived from A/Hong Kong/4801/2014, and six pPol I plasmids encoding the remaining six segments of high-yield A/Puerto Rico/8/34 virus[51] were used. These eight pPol I plasmids along with protein expression plasmids for PB2, PB1, PA, and NP derived from A/Puerto Rico/8/34 were transfected into 293T cells by use of TransIT 293 (Mirus) according to the manufacturer's instructions. At 48 h post-transfection, the supernatants were harvested and inoculated into hCK cells. The rescued viruses were sequenced to ensure the absence of unwanted mutations. Primer sequences are available upon request.

### Hybridoma generation
PBMCs were isolated from blood (30 ml) obtained from three volunteers, who were infected with the A/H1N1pdm09 virus in the 2015–2016 influenza season, by using Ficoll Paque Plus (GE Healthcare). The hybridomas that were generated by a fusion of PBMCs with SPYMEG cells were cultured and biologically cloned, as described previously[52]. For the screening, we performed an enzyme-linked immunosorbent assay (ELISA) using the culture media of hybridomas and purified A/Yokohama/94/2015 as an antigen, as described below.

### Construction of NA expression plasmids
Full-length NA was PCR amplified from viral cDNA or synthesized DNA and was cloned into the mammalian expression vector pCAGGS. All plasmids were sequenced to ensure the absence of unwanted mutations.

### Preparation of NA-VLPs
293T cells were transfected with pCAGGS encoding ebolavirus matrix protein VP40 with or without pCAGGS encoding the indicated NA by using TransIT 293. After 2 days, the supernatant of the transfected cells was harvested and ultra-centrifuged at 28,000 rpm with a 20% sucrose

cushion. The pellet containing VP40-induced VLPs displaying NA (NA-VLPs) was resuspended in PBS and stored at −80 °C.

## ELISA

ELISA plates (96-well) were coated with 10 µg/ml of the purified virus or the NA-VLPs at 4 °C. After being blocked with five times-diluted Blocking One (Nakarai), these plates were incubated with the culture medium of the hybridomas or four-fold diluted mAb (50–0.003 µg/ml). A horseradish peroxidase (HRP)-conjugated goat anti-human IgG, Fcγ Fragment-specific antibody (Jackson Immuno-Research) was used as the secondary antibody (1:5000 dilution), and the signal was developed using 3,3′,5,5′-tetramethylbenzidine (TMB) (Thermo Fisher Scientific) as the substrate. The reaction was stopped with 2 N sulfuric acid, and $OD_{450}$ values were then measured by using Softmax Pro, v5.4.5 (Molecular Devices). The mean $OD_{450}$ at each dilution was obtained by subtracting the $OD_{450}$ value for the control well (coated with VLP, not including NA). The minimal binding concentrations of the mAbs were determined as the lowest concentration providing an $OD_{450}$ value of >0.1.

## Expression and purification of monoclonal human or mouse IgG

Total RNA was extracted from the hybridomas by using an ISOGEN (Nippon gene). The VH and VL sequences of the antibodies were cloned into the expression vector Mammalian PowerExpress System (TOYOBO), together with the human or mouse constant gamma heavy (IgG1 or IgG2a) and kappa light chain coding sequences. Monoclonal antibodies were expressed by Expi293F cells and purified by using a protein A column and the automated chromatography system ÄKTA pure 25 (GE Healthcare)[53]. The germline reverted antibody was produced based on the germline sequence estimated in IgBlast software (https://www.ncbi.nlm.nih.gov/igblast/) with intact CDR H3 and L3 sequences.

## ELLA

The inhibition of NA activity was measured by using an enzyme-linked lectin assay (ELLA), as described previously[54]. Briefly, four-fold diluted antibodies (50–0.01 µg/ml) were mixed with a pre-determined amount of NA-VLPs diluted in PBS containing 1% BSA and 0.05% Tween 20 (PBS-T). The concentrations of NA-VLPs were determined based on the NA activity of each NA-VLP. The mixture was transferred to 96-well plates coated with fetuin (Sigma) and then incubated for 18 h at 37 °C. The plates were washed with PBS-T and then peroxidase-conjugated peanut agglutinin (PNA; Sigma) was added to detect galactose exposed by removal of the sialic acids on fetuin. The plates were incubated at room temperature for 2 h in the dark and then washed with PBS-T before the addition of o-phenylenediamine dihydrochloride (OPD) substrate (Sigma). The reaction was stopped by the addition of 1 N sulfuric acid and $OD_{490}$ values were read by using Softmax Pro, v5.4.5 (Molecular Devices). The relative NA inhibition (NI) activity was calculated by dividing the $OD_{490}$ value of the test well by the $OD_{490}$ value of the "VLP only" well and multiplying by 100. The 50% inhibitory concentration ($IC_{50}$) was determined by nonlinear regression analysis (GraphPad Prism software).

## NA-Star assay

The NA-Star Influenza Neuraminidase Inhibitor Resistance Detection Kit (Applied Biosystems) was used to measure the ability of the anti-NA mAbs to inhibit the ability of viral NA to cleave a small, soluble chemiluminescent substrate. The assay was performed according to the manufacturer's protocol. In brief, 25 µl of four-fold diluted antibodies (50–0.01 µg/ml) was transferred to a white, flat bottom 96-well plate. Then, 25 µl of NA-displaying VLPs was added to each well and the plate was shaken and incubated for 20 min at 37 °C. The NA-Star substrate was prepared shortly before use and 10 µl of the substrate was added

to all wells. The plate was then incubated for 30 min at room temperature, and 60 µl of NA-Star accelerator solution was added to all wells immediately before the plate was read by using a microtiter plate reader (Turner BioSystems). The $IC_{50}$ value was determined by nonlinear regression analysis (GraphPad Prism software).

## In vitro microneutralization (MN) assay

To assess the neutralization potency of the mAbs, 100 $TCID_{50}$ (50% tissue culture infectious dose) of each indicated virus in MEM containing 0.3% bovine serum albumin (BSA-MEM) was incubated with two-fold diluted antibodies (50–0.78 µg/ml) at 37 °C for 30 min. hCK cells were washed with BSA-MEM and then incubated with the antibody–virus mixture in quadruplicate at 37 °C for 1 h. After incubation, BSA-MEM containing 1 µg/ml ʟ-(tosylamido-2-phenyl) ethyl chloromethyl ketone (TPCK)-treated trypsin was added to cells, and the plates were incubated for 3 days at 37 °C before the cytopathic effect (CPE) was examined. Antibody titers required to reduce virus replication by 50% ($IC_{50}$) were calculated by using the Reed and Muench method.

## In vivo protection test

Six-week-old female BALB/c or DBA2 mice (Japan SLC) for type A or type B were intraperitoneally inoculated with each indicated mAb at 10, 2, or 0.4 mg/kg in 250 µl of PBS. After 24 h, the mice were anesthetized with isoflurane and intranasally challenged with 10 $MLD_{50}$ (50% mouse lethal dose) of A/H1N1pdm09 virus (mouse-adapted A/California/04/2009), A/H5N1 virus (A/Vietnam/1203/2004), A/H3N2 virus (mouse-adapted A/Aichi/2/68), A/H7N9 virus (A/Anhui/1/2013), or B/Yamagata virus (B/Massachusetts/02/2012) in 50 µl of PBS. The body weights of four mice per group were monitored daily for 14 days. Mice that lost 25% or more of their initial body weight were scored as dead and euthanized according to institutional guidelines.

## Virus titers in lung

Four mice per group were intraperitoneally injected with the indicated antibodies at 10 mg/kg. One day later, the mice were challenged with 10 $MLD_{50}$ of the A/H1N1pdm09 virus. On days 2 and 4 post-infection, the mice were euthanized and the virus titers in the lungs were determined by using plaque assays in hCK cells.

## Selection of escape mutants

Escape mutants were selected by passaging A/H3N2 virus (A/Hong Kong/4801/2014) in the presence of DA03E17. Two-fold diluted antibodies (50–0.78 µg/ml) were incubated with 100- or 1000-fold diluted virus for 30 min at 37 °C. MDCK cells were washed with BSA-MEM and then incubated with the antibody–virus mixture at 37 °C for 1 h. After this incubation, BSA-MEM containing 1 µg/ml TPCK-treated trypsin was added to the cells and the plates were incubated for 3 days at 37 °C. Virus-containing supernatant was harvested from the CPE-positive well that contained the highest antibody concentration and was used for the next passage. We regarded a virus to be an escape mutant when it replicated well in the presence of mAb at 50 µg/ml.

## Competitive binding assay

ELISA plates were coated overnight at 4 °C with 10 µg/ml of the VLPs. After being blocked with five times-diluted Blocking One, these plates were incubated for 1 h with four-fold diluted DA03E17 or 1G01 (100–0.006 µg/ml) that possessed the Fc region of mouse IgG2a. The plates were then washed with PBS-T, followed by the addition of individual human IgG1 antibodies (DA03E27, 1G01, or DA05A30) at concentrations adjusted to achieve an $OD_{450}$ value of 1.5. After 1 h, an HRP-conjugated goat anti-human IgG, Fcγ Fragment-specific antibody (Jackson Immuno-Research) was used as the secondary antibody (1:5000 dilution). The plates were incubated for 1 h and then washed with PBS-T before the addition of the TMB substrate. The reaction was

stopped with 2 N sulfuric acid, and $OD_{450}$ values were then measured. **$P < 0.01$ (two-way ANOVA followed by Dunnett's tests).

### Viral growth kinetics

Triplicate wells of confluent hCK cells were infected with the virus at a multiplicity of infection (MOI) of 0.0001 and incubated with BSA-MEM containing 1 μg/ml TPCK-treated trypsin at 37 °C. Supernatants were harvested at 12, 24, 48, and 72 h post-infection. Virus titers were determined by the use of plaque assays with hCK cells.

### Structural analysis

Amino acid positions were plotted on the crystal structure of the NA protein of A/Minnesota/11/2010 (A/H3N2) (PDB accession code: 6N6B) by using the PyMOL molecular graphics system.

### Statistical analysis

Two-way analysis of variance (ANOVA) followed by Dunnett's test and the log-rank test were performed using GraphPad Prism software. The exact $p$-values for each experiment are listed in the Source Data file. $P$ values < 0.01 were considered significantly different. No samples were excluded from the analysis.

### Reporting summary

Further information on research design is available in the Nature Portfolio Reporting Summary linked to this article.

## Data availability

All data analyzed during this study are included in this article. The raw data generated in this study are provided in the Source Data file. Antibody sequences have been deposited in GenBank [accession numbers: OP311729 (HP02A67 Heavy chain), OP311734 (HP02A67 Light chain), OP311730 (DA03E17 Heavy chain), OP311732 (DA03E17 Light chain), OP311731 (DA05A30 Heavy chain), and OP311733 (DA05A30 Light chain)]. The NA sequences of human isolates were collected from the GISAID database (https://www.gisaid.org/). Source data are provided with this paper.

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

## Acknowledgements
We thank Dr. Kiyoko Iwatsuki-Horimoto for assistance with experiments; Drs. Chiharu Kawakami, Emi Takashita, and Setsuko Nakajima for providing us with influenza viruses; and Dr. Susan Watson for editing the manuscript. This work was supported by a Japan Program for Infectious Diseases Research and Infrastructure [JP21wm0125002] from the Japan Agency for Medical Research and Development (AMED), JSPS KAKENHI [Grant Numbers 18K07141 and 21K07038], the National Institutes of Allergy and Infectious Diseases funded Center for Research on Influenza Pathogenesis [CRIP; HHSN272201400008C], Center for Research on Influenza Pathogenesis and Transmission [75N93021C00014], and the MUFJ-FG Vaccine Development Support Project [Code 215100000358].

## Author contributions
A.Y., S.Y., and Y.K. designed the study, analyzed the data, and wrote the manuscript. A.Y., S.Y., M.K., Y.S.-T., and M.O. performed the experiments. All authors reviewed and approved the manuscript.

## Competing interests
Y.K. has received speaker's honoraria from Toyama Chemical and Astellas Inc.; has received grant support from Daiichi Sankyo Pharmaceutical, Toyama Chemical, Tauns Laboratories, Inc., Otsuka Pharmaceutical Co., Ltd., Shionogi & Co. LTD KM Biologics, Kyoritsu Seiyaku, Shinya Corporation, and Fuji Reb; and is a co-founder of FluGen. The other authors declare no competing interests.
