## [Peer Review File · Nature Communications]

A broadly protective human monoclonal antibody targeting the sialidase activity of influenza A and B virus neuraminidasesReviewers' Comments:

Reviewer #1:

Remarks to the Author:

The manuscript describes the discovery and characterization of a novel human broadly cross-reactive neuraminidase (NA)-directed monoclonal antibody (mAb) that neutralizes and confers protections in mice from several subtypes of influenza A as well as influenza B viruses. The authors characterized three mAbs from a patient with a pandemic H1N1 infection by using hybridoma cells described in their earlier paper. Amongst the three mAbs, the authors found that the antibody DA03E17 had the broadest reactivity and targeted the enzymatic active site on NA. It was also worth noting that the authors used the best-in-class broadly cross-reactive anti-NA antibody 1G01 from the literature as a comparator/benchmark for their mAbs throughout the study, which I found very helpful. To my knowledge, DA03E17 is the second antibody that show exceptional broad cross-reactivity and protective efficacy against a wide range of influenza A and influenza B viruses. It is particularly important that this type of antibodies is now found in two independent subjects with different influenza virus exposure/infection history, and isolated by two independent research groups by different approaches. The immunogenetic composition of DA03E17 appeared to be different from that of 1G01, underscoring at least two non-redundant pathways for achieving the exceptionally broad NA cross-reactivity existed in human immunoglobulin repertoire. While the manuscript clearly shows premise, there are some weaknesses in their experimental design and data interpretation that should be addressed.

Major comments:

1. Line 75: The epitope of DA03E17 had not been determined by the present study. Based on the specificity and the positions of escape mutations, the antibody footprint of DA03E17 would likely overlap with that of 1G01.
2. Line 126: The authors concluded that the DA03E17 showed broader reactivity and more potent inhibition than 1G01. It was entirely based on the subtypes and strains used in this study, and there are many more NA subtypes that are tested in Stadlbauer et al. 2019 to characterize the 1G01. It does not seem fair to make that statement when the authors compared the two antibodies by using a substantially smaller (and potentially selective) panel of NAs.
3. Line 134: The authors stated "...DA03E17 protected 100%, 75%, 25% or 25% of mice...". This is misleading since only 4 mice per group were used. The resolution (one unit) of these experiments is 25% which is severely underpowered to compare groups. Did the authors perform sample size and power calculation prior to these experiments? Even if the experiments were repeated (I could not find the info) N = 4 might not have enough power to statistically distinguish one from others.
4. Regarding the passive transfer study, do the authors have any plausible explanation for why 1G01 did not confer protection against H5N1 and H7N9 challenge? Stadlbauer et al. 2019 showed full protection with 1G01 at 5mg per kg with minimal weight loss.
5. Figure 1: It is of great interest to know if the DA03E17 (and 1G01) can bind more recent H3N2 virus, specifically viruses after 2014 that acquired a new glycosylation site near the catalytic pocket.
6. Table 4: It would be nice to have accompanied enzymatic activity and viral fitness/growth kinetics of the viruses with mutated NA. It is particularly interesting whether the D151X mutations have deleterious effect on virus growth.
7. Related to the point above, do the NA mutant viruses have different sensitivity to NA inhibitors?
8. Line 190: "...there have been few reports of broadly reactive antibodies against NA (29)." Is there any other reports since the authors mentioned "few" in the sentence?
9. Line 196: "Furthermore, DA03E17 showed broader cross-reactivity and better neutralization capability than 1G01, an NI mAb previously reported to have cross-reactivity against IAVs and IBVs, and protected mice from lethal challenges with several subtypes of IAVs, suggesting that DA03E17 has potential as a therapeutic agent against seasonal IAVs and IBVs." First, this sentence is extremely long. Second, I disagree with this statement as the authors did not test antibodies against a large enough panel to make the judgement.

Minor comments:

1. Line 87: Please provide some stats. How many clones the authors screened, etc.
2. Line 100: Please add reference for 1430E3/9 antibody.
3. Figure 3B: It would be helpful to color code residues (letters) according to their chemical properties and to include some more NA subtypes in the alignment.
4. Line 175: Please elaborate how the inferred germline sequence is generated. Are the CDR H3 and L3 intact?
5. Line 285: Please elaborate how the SPYMEG cells were cultured and biologically.
6. Line 289: Preparation of NA-VLPs. How were the VLPs characterized and quantified? Were they tested for enzymatic activity?
7. Line 306: Please elaborate how the human IgG were expressed and purified. Providing sequences (not only CDRs) would also be great for the scientific community.

Reviewer #2:

Remarks to the Author:

In the manuscript, Yusuhara et al. described the isolation and characterization of anti-NA human antibodies against influenza virus, among which DA03E17 was identified as a broadly reactive antibody against influenza A and influenza B viruses. The research team focused on DA03E17 and tested the binding activities to NA-VLPs and the inhibition of NA sialidase activity of selected subtypes of IVAs and IVBs. The prophylactic effect of DA03E17 was evaluated in mouse model. The authors also selected escape mutants by passaging A/H3N2 virus in the presence of DA03E17 antibody and identified the highly conserved D151 as the key epitope amino acid. Lastly, the germline antibody of DA03E17 has been prepared to investigate the maturation of this antibody and suggested that the broad activity was obtained through somatic hypermutation. Overall, the identification of a broadly neutralizing NA antibody and the characterization of its key epitope and the mechanism of action residues are important for understanding NA antibody and developing broadly effective agent for combating both influenza A and influenza B. Although the fine identification of escape mutants revealed a piece of epitope information, the key evidence is missing. Structural approaches, for example, X-ray crystallography or cry-EM should be used to determine how DA03E17 binds to NAs of both IAVs and IBV.

1. According to the manuscript, D151 is the key epitope amino acid, which is highly conserved among IAVs and IBV, while the other key amino acid T439 only presented in IAVs but not IBVs. How to explain this finding? Perhaps the binding patterns of DA03E17 to IAVs and IBVs are different?
2. An antibody named Z2B3 which is also a broad neutralizing antibody binds within the conserved active site of the neuraminidase protein (PMC7542365). Other studies also highlighted the active site as the epitope of neutralizing antibodies. The authors should specify the novelty of the epitope or working mechanism of their antibody.
3. For the escape mutant experiment, did the authors have the observation that the infectivity of NA D151 mutated viruses were different from the wild type of virus? Since the NA active site includes D151, the mutations on this site may influence the virulence.
4. The antibodies were isolated from patients who were infected with A/H1N1pdm09 virus. The authors also hypothesized that "affinity maturation through somatic hypermutations in the VH and VL genes play an essential role in the high binding affinity and broad reactivity of DA03E17." The broad neutralizing may be acquired by the repeated exposure of similar but slightly different NAs. Do the patients have a known history of infection or vaccination other than A/H1N1pdm09?

Reviewer #3:

Remarks to the Author:

Periodic antigenic drift has often allowed influenza A and B viruses to evade vaccine-elicited responses and resist therapeutic treatment. Towards the development of vaccine or therapeutics that has broad inhibitory activity and protection, much work has focused on identifying highly conserved epitopes on the hemagglutinin, the chief target of the neutralization antibody response. However, there is an alternative viral protein, the neuraminidase, which can confer a separate and independent correlate of protection against disease. Here, Yasuhara and colleagues describe a human monoclonal antibody that can bind and inhibit enzyme activity of neuraminidases from a panel of influenza A and influenza B viruses. While in vitro data demonstrate that DA03E17 has broad activity against both groups of viruses – these activities may not correlate with protection in vivo. The manuscript is straightforward and easy to understand. To strengthen the claims of broad activity a number of additional experiments or analyses should be addressed prior to publication.

The manuscript is straightforward and well-written. This is arguably the second group to describe a human mAb that targets the influenza A and B virus neuraminidases, highlighting antibodies can recognize regions that are conserved in the neuraminidases of the two viral groups. This can provide further impetus in providing support of incorporating neuraminidase in the design of broad-spectrum vaccines. In vitro assays support the claim that DA03E17 has broad binding and antiviral activity when compared 1G01 and has great potential as a prophylactic.

However, while data in Figure 1 demonstrate the larger breadth of activity of DA03E17 compared to 1G01 – this is based off of the chosen viruses used by the authors, which can provide a misleading notion that indeed DA03E17 has broader range. Stadlbauer et al (PMID: 31649200) demonstrate that 1G01 can bind and has ELLA activity against N1, N2, N4, N5, N6, N7, N8, N9 and B NAs. DA03E17 is only shown to have activity against N1, N2, N9 and several B NAs. Authors could provide more evidence that DA03E17 has as much or if not more range that what has been published. Or perhaps discuss their rationale for their hypothesis. Of note, in lines 200 to 201, the authors claim that they hypothesize that DA03E17 can recognize N3, N4, N5, N6, N7 or N8 NAs because DA03E17 can bind to B NA – can this be clarified? While the D in 151 is conserved between the NA of A and B viruses, the T in 439 is not.

While the ELLA and Star activity assays are well-established methods to measure inhibition of enzyme activity, it may increase the rigor of the study overall if the authors could determine the impact of enzyme inhibition by measuring viral egress or spread in an in vitro assay?

In Figure 3 and Supplementary Figure 1, it is a bit odd that at a lower dose of mAb (S. Figure 1) during H3 or H7 challenge – more mice survived, which is counter intuitive. Could the authors please clarify or discuss these results? Perhaps it is due to the small number of animals per group – which should also be discussed or addressed.

While in vitro assays demonstrate broad activity against A and B viruses, authors should at least demonstrate some protection experiments against B viruses to strengthen the claim. Moreover, viral lung titers are now shown against any of the IAV challenge experiments as this can clearly demonstrate that protection from disease and death is due to the mAb delaying or inhibiting viral egress in vivo.

Reviewer #1:

Major comments:

1. Line 75: The epitope of DA03E17 had not been determined by the present study. Based on the specificity and the positions of escape mutations, the antibody footprint of DA03E17 would likely overlap with that of 1G01.

In response to the reviewer's comment, we performed a competitive binding assay to evaluate whether the epitope of DA03E17 overlapped with that of 1G01. Our results suggest that the epitopes of DA03E17 and 1G01 overlap but are not completely. We present these data as Figure S3, and describe them in the text (Page 12, lines 183-196).

2. Line 126: The authors concluded that the DA03E17 showed broader reactivity and more potent inhibition than 1G01. It was entirely based on the subtypes and strains used in this study, and there are many more NA subtypes that are tested in Stadlbauer et al. 2019 to characterize the 1G01. It does not seem fair to make that statement when the authors compared the two antibodies by using a substantially smaller (and potentially selective) panel of NAs.

We additionally investigated the reactivity of DA03E17 against N4-, N5-, N8-, N6-, or N7-NA, which were used in Stadlbauer et al. 2019. DA03E17 was found to bind to all of these additional NAs (See Fig. 1B), indicating that our antibody has broader reactivity than 1G01. We present these data as Figure 1B,

Table S2, and Table S3, and describe them in the text (Page 7, lines 95-100; Page 8, line 102).

3. Line 134: The authors stated "...DA03E17 protected 100%, 75%, 25% or 25% of mice...". This is misleading since only 4 mice per group were used. The resolution (one unit) of these experiments is 25% which is severely underpowered to compare groups. Did the authors perform sample size and power calculation prior to these experiments? Even if the experiments were repeated (I could not find the info) N = 4 might not have enough power to statistically distinguish one from others.

As the reviewer points out, it is difficult to make statistical comparisons with N = 4. To avoid any confusion, we have avoided statistical comparisons in the mouse study (Page 10, lines 142-149).

4. Regarding the passive transfer study, do the authors have any plausible explanation for why 1G01 did not confer protection against H5N1 and H7N9 challenge? Stadlbauer et al. 2019 showed full protection with 1G01 at 5mg per kg with minimal weight loss.

We performed the mouse challenge test under slightly different conditions in terms of the virus strain and infectious dose from those used by Stadlbauer et al. 2019. Since there is no difference in the NA sequence of the virus strains used, the difference in the challenge dose (10 MLD₅₀ in our study vs 5 MLD₅₀ in Stadlbauer et al. 2019) may have affected the results.

5. Figure 1: It is of great interest to know if the DA03E17 (and 1G01) can bind more recent H3N2 virus,

specifically viruses after 2014 that acquired a new glycosylation site near the catalytic pocket.

In response to the reviewer's comment, we have now included NAs from two recent H3N2 viruses (A/Osaka/163/2017 and A/Myanmar/18MI025/2018) and performed ELISA and ELLA. DA03E17 recognized these strains possessing the N-linked glycosylation consensus sequence near the active site of NA and inhibited their sialidase activity. We present these data as Figure 1B, Table S2, and Table S3.

6. Table 4: It would be nice to have accompanied enzymatic activity and viral fitness/growth kinetics of the viruses with mutated NA. It is particularly interesting whether the D151X mutations have deleterious effect on virus growth.

In response to the reviewer's comment, we compared the growth kinetics of viruses possessing D151G or D151N in hCK cells. We found that mutant viruses possessing the mutations responsible for escape from DA03E17 (D151G or D151N) replicated to significantly lower titers than the wild-type virus. We present these data as Figure 4 and describe them in the text (Page 13, lines 203-207).

7. Related to the point above, do the NA mutant viruses have different sensitivity to NA inhibitors?

Some studies have shown that an amino acid substitution at 151 reduced sensitivities to NA inhibitors (Sheu et al. 2008, Mishin et al. 2014). We have added this information to the text (Page 17, lines 264-265).

8. Line 190: "...there have been few reports of broadly reactive antibodies against NA (29)." Is there any other reports since the authors mentioned "few" in the sentence?

Since no other papers have been reported, we corrected "few" to "one" (Page 15, lines 221-224).

9. Line 196: "Furthermore, DA03E17 showed broader cross-reactivity and better neutralization capability than 1G01, an NI mAb previously reported to have cross-reactivity against IAVs and IBVs, and protected mice from lethal challenges with several subtypes of IAVs, suggesting that DA03E17 has potential as a therapeutic agent against seasonal IAVs and IBVs." First, this sentence is extremely long. Second, I disagree with this statement as the authors did not test antibodies against a large enough panel to make the judgement.

As commented in the above response, we examined the binding and NI activities of DA03E17 against N4-, N5-, N8-, N6-, or N7-NA, which were used in Stadlbauer et al. 2019 (Figure 1B, Table S2 and Table S3). These results indicate that our antibody has broader reactivity than 1G01. We have also shortened the sentence (Page 15, lines 229-237).

Minor comments:

1. Line 87: Please provide some stats. How many clones the authors screened, etc.

In response to the reviewer's comment, we now describe the number of screened clones in the text (Page 7, line 90).

2. Line100: Please add reference for 1430E3/9 antibody.

We added a reference for anti-B-HA mAb 1430E3/9 to the text (Page 8, line 104).

3. Figure 3B: It would be helpful to color code residues (letters) according to their chemical properties and to include some more NA subtypes in the alignment.

We modified the figure to show the chemical properties of the amino acids (Figure 3B).

4. Line 175: Please elaborate how the inferred germline sequence is generated. Are the CDR H3 and L3 intact?

The germline sequence was estimated in IgBlast software, and CDR H3 and L3 were not modified from the original DA03E17. We have added this information to the text (Page 22, lines 340-347).

5. Line 285: Please elaborate how the SPYMEG cells were cultured and biologically.

We have included the culture conditions for the SPYMEG cells in the text (Page 19, line 289).

6. Line 289: Preparation of NA-VLPs. How were the VLPs characterized and quantified? Were they tested for enzymatic activity?

Each time we prepared NA-VLPs, their neuraminidase activities were measured in the ELLA. The

concentrations of the NA-VLPs used in the experiments were determined based on the NA activity of each

NA-VLP. We have added this point to the text (Page 22, line 352).

7. Line 306: Please elaborate how the human IgG were expressed and purified. Providing sequences (not only CDRs) would also be great for the scientific community.

We have added detail methods for IgG expression to the text (Page 22, lines 337-344), and deposited the antibody sequences in Genbank (Page 27, lines 430-433).

Reviewer #2:

Although the fine identification of escape mutants revealed a piece of epitope information, the key evidence is missing. Structural approaches, for example, X-ray crystallography or cry-EM should be used to determine how DA03E17 binds to NAs of both IAVs and IBV.

We would like to thank the reviewer for this suggestion. For the next steps, we will collaborate with an X-ray crystallographer to determine the co-crystal structure of each antibody together with NA to explore the exact binding mode.

1. According to the manuscript, D151 is the key epitope amino acid, which is highly conserved among IAVs and IBV, while the other key amino acid T439 only presented in IAVs but not IBVs. How to explain this finding? Perhaps the binding patterns of DA03E17 to IAVs and IBVs are different?

It is possible that the key amino acids for antibody binding may differ between IAV-NA and IBV-NA; a future co-crystal experiment will address this issue.

2. An antibody named Z2B3 which is also a broad neutralizing antibody binds within the conserved active site of the neuraminidase protein (PMC7542365). Other studies also highlighted the active site as the epitope of neutralizing antibodies. The authors should specify the novelty of the epitope or working mechanism of their antibody.

Although previous studies have reported some monoclonal antibodies, including Z2B3, that possess cross-reactivity against viruses belonging to several subtypes of IAV or both lineages of IBV, there has been only one previous report of a human monoclonal antibody, 1G01, binding to both IAV-NAs and IBV-NAs. Our antibody has broader reactivity and likely a different footprint than 1G01. Therefore, our manuscript provides important information on the properties of broadly cross-reactive antibodies against NA.

3. For the escape mutant experiment, did the authors have the observation that the infectivity of NA D151 mutated viruses were different from the wild type of virus? Since the NA active site includes D151, the mutations on this site may influence the virulence.

In response to the reviewer's comment, we compared virus growth kinetics in hCK cells. We found that mutant viruses possessing the mutations responsible for escape from DA03E17 (D151G or D151N) replicated to significantly lower titers than the wild-type virus. We present these data as Figure 4 and describe them in the text (Page 13, lines 203-207).

4. The antibodies were isolated from patients who were infected with A/H1N1pdm09 virus. The authors also hypothesized that "affinity maturation through somatic hypermutations in the VH and VL genes play an essential role in the high binding affinity and broad reactivity of DA03E17." The broad neutralizing may be acquired by the repeated exposure of similar but slightly different NAs. Do the

patients have a known history of infection or vaccination other than A/H1N1pdm09?

We agree with reviewer's comments, but we could not get any information about the patients'

history of infection or vaccination.

Reviewer #3:

However, while data in Figure 1 demonstrate the larger breadth of activity of DA03E17 compared to 1G01 – this is based off of the chosen viruses used by the authors, which can provide a misleading notion that indeed DA03E17 has broader range. Stadlbauer et al (PMID: 31649200) demonstrate that 1G01 can bind and has ELLA activity against N1, N2, N4, N5, N6, N7, N8, N9 and B NAs. DA03E17 is only shown to have activity against N1, N2, N9 and several B NAs. Authors could provide more evidence that at DA03E17 has as much or if not more range that what has been published. Or perhaps discuss their rationale for their hypothesis. Of note, in lines 200 to 201, the authors claim that they hypothesize that DA03E17 can recognize N3, N4, N5, N6, N7 or N8 NAs because DA03E17 can bind to B NA – can this be clarified?

In response to the reviewer’s comment, we investigated the reactivity of DA03E17 against N4-, N5-, N8-, N6-, or N7-NA, which were used in Stadlbauer et al. 2019. DA03E17 was found to bind to all of the additional NAs, indicating that our antibody has broader reactivity than 1G01. We present these data as Figure 1B, Table S2, and Table S3, and describe them in the text (Page 7, lines 95-100; Page 8, line 102)

While the ELLA and Star activity assays are well-established methods to measure inhibition of enzyme activity, it may increase the rigor of the study overall if the authors could determine the impact of enzyme inhibition by measuring viral egress or spread in an in vitro assay?

To evaluate the in vitro neutralizing potency of our antibodies, we performed a microneutralization assay using authentic viruses (see 'MN' in Figure 1A). These results indicate that our DA03E17 inhibited virus spread in vitro.

In Figure 3 and Supplementary Figure 1, it is a bit odd that at a lower dose of mAb (S. Figure 1) during H3 or H7 challenge – more mice survived, which is counter intuitive. Could the authors please clarify or discuss these results? Perhaps it is due to the small number of animals per group – which should also be discussed or addressed.

As the reviewer points out, the unexpected results in the mouse experiments are likely due to the small sample size. To avoid any confusion, we have revised the wording, avoiding statistical comparisons, in the discussion of the mouse study (Page 10, lines 142-149).

While in vitro assays demonstrate broad activity against A and B viruses, authors should at least demonstrate some protection experiments against B viruses to strengthen the claim. Moreover, viral lung titers are now shown against any of the IAV challenge experiments as this can clearly demonstrate that protection from disease and death is due to the mAb delaying or inhibiting viral egress in vivo.

In response to the reviewer's comment, we examined the in vivo protective efficacy of DA03E17 against B/Yamagata virus. Body weight changes of infected mice revealed that DA03E17 exhibited a level of protection against B/Yamagata virus similar to that achieved by 1G01. We present these data as Figures

2A and S2 and describe them in the text (Page 10, lines 142-149).

We also assessed virus titers in the lungs of A/H1N1pdm09-infected mice that received each mAb at 10 mg/kg. On days 2 and 4 post-infection, the virus titers in the lungs of the mice that received DA03E17 were significantly lower than those in the lungs of the mice that received the negative control 1430E3/9.

These results demonstrate that DA03E17 suppresses virus propagation in vivo. We present these data as

Figures 2B and describe them in the text (Page 10, lines 152-157).

Reviewers' Comments:

Reviewer #1:

Remarks to the Author:

The authors had addressed many of my concerns and questions. The manuscript has been greatly improved and strengthened. The authors had now convincingly shown that the DA03E17 targets the NA catalytic site and substantially cross-competes with the 1G01. It is also worth emphasizing that the DA03E17 is the second reported exceptionally broadly cross-protective NA antibody which is isolated from a different individual and uses different IGHV and IGLV genes from the 1G01. These facts provide new insights into the NA-based vaccine efforts targeting this site of vulnerability.

Regarding my previous comment about the reactivity against the NAs derived from contemporary H3N2 viruses that possess an additional glycan at position 245, it was a bit surprising that both DA03E17 and 1G01 did not show reduced activity against those viruses (Fig. 1B). Because the residue 245 is part of the 1G01 epitope, I expected somewhat lower reactivity to the NAs with glycan-245 for 1G01 (and DA03E17 to some degree). Another related NA catalytic site antibody B10 which has quite similar epitope to 1G01 showed much reduced activity against the glycan-245 containing viruses (Wan et al. Nat Microbiol. 2019). Although it might be due to the difference in assays, e.g., glycan occupancy in NA-VLP vs. egg-propagated viruses, etc., the authors may want to note this antigenic drift which may or may not alter antigenicity of the NA catalytic site.

Reviewer #3:

Remarks to the Author:

The authors have addressed concerns raised by the referees.

Reviewer #4:

Remarks to the Author:

Co-reviewer with reviewer #1

Reviewer #1:

Regarding my previous comment about the reactivity against the NAs derived from contemporary H3N2 viruses that possess an additional glycan at position 245, it was a bit surprising that both DA03E17 and 1G01 did not show reduced activity against those viruses (Fig. 1B). Because the residue 245 is part of the 1G01 epitope, I expected somewhat lower reactivity to the NAs with glycan-245 for 1G01 (and DA03E17 to some degree). Another related NA catalytic site antibody B10 which has quite similar epitope to 1G01 showed much reduced activity against the glycan-245 containing viruses (Wan et al. Nat Microbiol. 2019). Although it might be due to the difference in assays, e.g., glycan occupancy in NA-VLP vs. egg-propagated viruses, etc., the authors may want to note this antigenic drift which may or may not alter antigenicity of the NA catalytic site.

In response to the reviewer's comment, we now describe this point in the main text (page 17, lines

270–283).